# *PDE3A* and *GSK3B* as Atrial Fibrillation Susceptibility Genes in the Chinese Population via Bioinformatics and Genome-Wide Association Analysis

**DOI:** 10.3390/biomedicines11030908

**Published:** 2023-03-15

**Authors:** Zechen Zhou, Yu Wang, Xiaoyi Li, Yinan Zhang, Lichuang Yuan, Dafang Chen, Xuedong Wang

**Affiliations:** 1Department of Epidemiology and Biostatistics, School of Public Health, Peking University Health Science Center, Beijing 100191, China; 2Department of Cardiology, Beijing Hepingli Hospital, Beijing 100013, China

**Keywords:** atrial fibrillation, differentially expressed genes, pathway gene network, GWAS analysis

## Abstract

Background: Atrial fibrillation (AF) is the most common cardiac arrhythmia, with uncovered genetic etiology and pathogenesis. We aimed to screen out AF susceptibility genes with potential pathogenesis significance in the Chinese population. Methods: Differentially expressed genes (DEGs) were screened by the Limma package in three GEO data sets of atrial tissue. AF-related genes were identified by combination of DEGs and public GWAS susceptibility genes. Potential drug target genes were selected using the DrugBank, STITCH and TCMSP databases. Pathway enrichment analyses of AF-related genes were performed using the databases GO and KEGG databases. The pathway gene network was visualized by Cytoscape software to identify gene–gene interactions and hub genes. GWAS analysis of 110 cases of AF and 1201 controls was carried out through a genome-wide efficient mixed model in the Fangshan population to verify the results of bioinformatic analysis. Results: A total of 3173 DEGs were identified, 57 of which were found to be significantly associated with of AF in public GWAS results. A total of 75 AF-related genes were found to be potential therapeutic targets. Pathway enrichment analysis selected 79 significant pathways and classified them into 7 major pathway networks. A total of 35 hub genes were selected from the pathway networks. GWAS analysis identified 126 AF-associated loci. *PDE3A* and *GSK3B* were found to be overlapping genes between bioinformatic analysis and GWAS analysis. Conclusions: We screened out several pivotal genes and pathways involved in AF pathogenesis. Among them, *PDE3A* and *GSK3B* were significantly associated with the risk of AF in the Chinese population. Our study provided new insights into the mechanisms of action of AF.

## 1. Introduction

Atrial fibrillation (AF) is the most common nonbenign cardiac arrhythmia in clinical practice and is one of the leading causes of stroke, heart failure, cardiovascular disease, and sudden death [1,2]. With the increasing age of populations, AF incidence is increasing rapidly worldwide [3]. The strongest risk factor for AF is old age, along with gender, smoking, alcohol consumption, body mass index, hypertension, left ventricular hypertrophy, significant heart murmur, heart failure, and myocardial infarction [4,5]. However, the etiology and pathophysiologic mechanisms of AF are incompletely understood. In addition, new drugs specially designed for the therapy of AF remain suboptimal, and patients have to depend on antique antiarrhythmic drugs, such as amiodarone, sotalol, propafenone, and flecainide, which have limited efficacy and significant side effects [6]. 

Recently, numerous studies have suggested that AF cases in the general population have a significant genetic component, even beyond traditional risk factors [7]. Genome-wide association studies (GWAS) have been applied to more than 30 million individuals worldwide and have discovered more than 100 distinct genetic loci associated with AF [8,9]. However, previous studies have had a variety of limitations. Combined with the results of the genetic studies thus far, genetic variation only accounts for approximately 40–50% of the heritability for AF. The sample size and ethnic diversity of GWASs have become increasingly large, but reproducibility of the identified association signals has become a serious challenge. The effects of many SNPs are difficult to verify in different ethnical populations. In addition, more than 95% of these GWAS variants are localized in noncoding regions, which may act through the effect of gene expression or pathway regulation. The main reason for these limitations is that conventional GWASs have only been able to identify SNPs that are associated with AF rather than causative, and there is always a lack of explanation for the biological significance behind the association.

Common approaches to further investigate the potential mechanisms of AF in a specific population are to combine the results of GWAS analysis with multiomics analysis of human atrial tissue by bioinformatics mining, including differentially expressed genes (DEGs) analysis, drug–gene interactions (DGIs) analysis, and pathway enrichment analysis. Bioinformatic analysis and studies using microarrays to measure gene expression can be employed to screen molecular markers in patients and healthy individuals. Microarray studies are commonly used to obtain gene expression profiles to uncover the pathogenesis of complicated diseases and for biomarker identification. In the last decade, a number of studies have tried to determine the transcriptomic changes in both AF patients and animal models using microarray technologies [10,11]. Several key pathways related to microRNAs (miRNAs), such as Ca^2+^-dependent signaling pathways, inflammatory and immune pathways, and apoptotic and cycle pathways, have been found [12]. Nevertheless, the results of bioinformatic analysis often lack the verification of real-world people.

In this study, we first integrated the multilevel biological information resources of AF and screened the set of candidate AF genes of biological significance based on the differentially expressed genes, the reported GWAS susceptibility genes, and the potential drug target genes. At the preliminary stage, the union of genes is extracted from the results of multiple data sets at the same molecular level, that is, as long as the results have been reported at least once, they are considered candidate AF-related genes. Then, we screened out the hub genes of AF through pathway enrichment analysis and the construction of the gene–gene interaction network of the candidate genes. We further carried out a GWAS analysis to verify the association between loci discovered by bioinformatics and AF in the Chinese population. Our study may be helpful in revealing the genetic etiology and pathogenesis underlying AF.

## 2. Materials and Methods

### 2.1. Identification of DEGs and Susceptibility Genes

Gene-expression profiles of AF were collected from the GEO database (www.ncbi.nlm.nih.gov/geo (accessed on 7 August 2022)) [13]. The GSE2240 data set includes the atrial myocardium tissues of samples from 10 patients with AF and 20 controls with sinus rhythm. The GSE128188 data set includes left and right atrial appendage tissues of samples from 5 AF patients and 5 controls with sinus rhythm. The GSE115574 data set includes the atrial tissues of samples from 15 patients with AF and 15 controls with sinus rhythm. All data in the present study were collected from public databases, so ethical approval from our institution was not needed.

The DEGs between patients with AF and controls with sinus rhythm were screened using the Linear Models for Microarray Data (LIMMA, http://www.bioconductor.org/packages/release/bioc/html/limma.html (accessed on 27 April 2022, verison 3.52.4, The Walter and Eliza Hall Institute of Medical Research, Melbourne, Australia).) package in R 4.2.1(R Foundation for Statistical Computing, Vienna, Austria) [14]. *p*-values < 0.05 after Bonferroni correction were chosen as cut-off criteria. Gene expression values of | log_2_ (fold change, FC) | > 0 were labeled as upregulated genes, and values < 0 were labeled as downregulated genes. Then, we obtained the list of AF genetic susceptibility sites and their mapping genes from the GWAS-catalog database and Open Targets database (accessed on 10 August 2022). Differentially expressed atrial fibrillation genes and susceptibility genes were combined as potential atrial fibrillation-related genes for subsequent analysis.

### 2.2. Identification of Potential Therapeutic Targets

The DrugBank database (https://go.drugbank.com/ (accessed on 13 August 2022)) [15] was used to search clinical drugs with indications for atrial fibrillation. The STITCH database (http://stitch.embl.de/ (accessed on 13 August 2022)) [16] was used to search the drug–gene interaction network of corresponding drugs and summarize the direct targets of drug action in the interaction network. The TCMSP database (https://old.tcmsp-e.com/index.php (accessed on 13 August 2022)) [17] was used to search the Chinese herbal medicines related to the treatment of atrial fibrillation and their pharmaceutical active ingredients. In order to include the effective components of traditional Chinese medicine for atrial fibrillation, only the active chemical components with set oral bioavailability (OB) greater than or equal to 30% and drug likeness (DL) value greater than or equal to 0.18 were incorporated. The collected active ingredients were used to find their targets in the TCMSP, and the targets were converted into the corresponding gene name in the UniProt database (https://www.uniprot.org (accessed on 13 August 2022)) [18].

### 2.3. Pathway Enrichment Analysis

GO enrichment analysis and KEGG enrichment analysis were performed on genes related to atrial fibrillation that could be potential therapeutic targets to explore the core functional pathways related to the pathogenesis and treatment of atrial fibrillation. *p*-value < 0.01 after Benjamini–Hochberg correction was used as the threshold to determine the enrichment effect of a gene subset in GO or KEGG entries. GO and KEGG enrichment analyses were performed with the R package clusterProfiler and enrichplot. The Cytoscape3.8.2 software (Cytoscape Consortium, California, USA) (ClueGO package verision 2.5.9, Laboratory of Integrative Cancer Immunology, Paris, France) [19] was used to construct pathway enrichment network map based on KEGG enrichment results to further screen the core functions or pathways related to the pathogenesis of atrial fibrillation.

### 2.4. Construction of Pathway Gene Network

The interaction relationships among gene targets under the pathway module were predicted through the String database (https://cn.string-db.org/ (accessed on 14 August 2022)) [20]. The score between nodes was set to 0.4, so that only the node interaction relationships greater than 0.4 can be included in the gene interaction network. Based on Cytoscape 3.8.2 software MCODE package (Verision 2.0.2, Gary Bader & Christian Lopes &Vuk Pavlovic, Toronto, Canada), the gene interaction network was constructed for each KEGG pathway network gene to mine the hub gene sets under each pathway network.

### 2.5. GWAS Study Design and Subjects

The GWAS analysis consisted of 110 cases of AF and 1201 controls free of AF from the Fangshan Family-based Ischemic Stroke Study in China (FISSIC) [21]. FISSIC is an ongoing community-based case-control genetic epidemiological study that started in June 2005, enrolling families in Fangshan District, a rural area located southwest of Beijing, China. The inclusion criteria for the subjects were as follows: (1) age older than 18 years at enrollment; (2) variables of sex, age, or AF condition not missing; and (3) subjects without single-gene hereditary disease or cancer. The diagnosis of atrial fibrillation was confirmed by a second-class or higher-class hospital. Atrial fibrillation (ICD-10 code I48) was defined as any event with a date of occurrence before the participant’s first visit for recruitment into the study. This study was approved by the Ethics Committee of the Peking University Health Science Center (Approval number: IRB00001052-13027), and written informed consent was provided by all participants.

### 2.6. Genotyping

DNA was extracted using a LabTurbo 496-Standard System (TAIGEN Bioscience Corporation, Taiwan, China). In addition, the purity and concentration of DNA were measured using ultraviolet spectrophotometry. Samples within the study were genotyped at the Capitalbio Technology Corporation using the Illumina ASA Chip. They were genotyped in 5 batches, grouped by origin of the samples, and with a balanced case-control mix on each array. Quality control (QC) [22] was performed on each sample, including >95% variant call rate, consistency between genotyped sex and the investigated sex, <3 SDs heterozygosity, consistency between IBDs and the investigated kinship, <5% Mendel errors, and no significant deviation from PCAs of ancestral background. QC was performed on each call set, including >95% sample call rate, Hardy–Weinberg equilibrium *p* > 1 × 10^−6^, minimum allele frequency (MAF) > 1%, and <10% Mendel errors. All QC was conducted using PLINK 1.9 software (https://www.cog-genomics.org/plink/ (accessed on 14 August 2022).

### 2.7. Statistical Analysis

Genome-wide association testing was performed using GEMMA (genome-wide efficient mixed model association) [23] software based on mixed effects model. The locus effect was decomposed into fixed effects on families and random effects on individuals by constructing the phylogenetic matrix, and the interindividual correlation within families was adjusted. Sex, age, and the first ten principal components were adjusted as covariates. To correct for multiple testing, a genome-wide significance threshold of *p* < 1 × 10^−8^ was performed. We inspected Manhattan plots and Q-Q plots for spurious associations and quantile–quantile plots to identify genomic inflation.

Analyses in addition to GEMMA were conducted using R version 4.2.1 (R Foundation for Statistical Computing, Vienna, Austria).

## 3. Results

### 3.1. Identification of DEGs and Susceptibility Genes

A total of 3173 DEGs were screened from the three GEO data sets: expression of 1757 genes was upregulated and the expression of 1432 genes was downregulated between AF patients and controls (Table 1). Among them, 22 genes were differentially expressed in all three data sets (Figure 1a). There were 16 genes with different expression regulatory directions in at least two data sets. The clusters of all DEGs are displayed in Figure 1a.

A total of 356 lists of AF genetic susceptibility sites and their mapping genes were obtained from the GWAS-Catalog database and Open Targets database. Among all the DEGs in atrial fibrillation (n = 3173), variants in 57 genes were found to be significantly associated with the risk of AF (*p* < 5 × 10^−8^). By combining GWAS and RNA expression information, 3472 genes were ultimately identified as potential AF-related genes, which are shown in Figure 1b.

### 3.2. Identification of Potential Therapeutic Targets

With atrial fibrillation as key words, 22 kinds of clinical drugs conforming to indications were searched based on the DrugBank database. A total of 137 direct targets were found in drug–gene interaction networks based on the STITCH database. A total of 439 herbal medicines related to the treatment of atrial fibrillation were found in the TCMSP database. Among them, 406 active ingredients were collected, which can be mapped to 278 targets. We intersected the targets with the potential AF-related genes. As a result, a total of 75 AF-related genes can be used as potential therapeutic targets, including 61 Chinese medicine targets and 26 chemical drug targets, as shown in Table 2 and Figure 2.

### 3.3. Pathway Enrichment Analysis

Pathway enrichment analysis was performed on the 75 AF-related genes above. GO enrichment analysis showed that 919 gene subsets were significantly enriched under specific biological processes. The first 20 biological processes with significant enrichment were selected to generate bubble maps (Figure 3a). In addition, KEGG enrichment analysis revealed significant enrichment of 79 gene subsets in specific pathways. The first 20 significantly enriched pathways were selected to generate the bubble map (Figure 3b).

### 3.4. Construction of Pathway Gene Network

GO enrichment network maps suggest clustering of 919 distinct biological processes into 80 functional categories. The main biological functions of AF-related genes mainly include the regulation of polysaccharides and transmembrane transporters, muscle contraction, and the negative feedback regulation process of catecholamine, glutaminergic compounds, and synaptic transmission. The pathway enrichment network constructed based on the KEGG enrichment results is shown in Figure 4. The nodes represent enriched biological processes or pathways. The node lines represent the number of common genes between biological processes or pathways, and the color indicates which functional group the node belongs to. According to shared genes among the pathways, 79 significant pathways were classified into 7 major pathway networks.

The gene interaction network was constructed for each KEGG pathway network gene to mine the hub gene set under each pathway network. The hub genes under the seven subnetworks included the oxidative stress and cellular signaling pathway, hormone regulation pathway, cell adhesion pathway, tumor inhibition pathway, cell cycle regulation pathway, lipid metabolism and inflammatory pathway, and mental illness-associated pathway (Table 3 and Figure 5). A total of 35 common hub susceptibility genes were obtained by constructing the pathway–gene network. Twenty-six of them belonged to the oxidative stress and cellular signaling pathways.

### 3.5. GWAS Analysis

A total of 496,798 genetic variants were tested after quality controls. Principal component analysis (PCA) revealed that participants in the present study are genetically East Asian, and there are no individuals who deviate significantly from their ancestral genetic background (Appendix A). The GWAS association analysis of AF in the Fangshan population revealed 126 AF-associated loci at genome-wide significance (*p* < 1 × 10^−8^) (Figure 6 and Table 4). The significance level accounts for multiple testing of independent variants with MAF ≥ 0.1% using a Bonferroni correction. *p* values (two-sided) were derived from a genome-wide efficient mixed model with the least-squares method.

### 3.6. Bioinformatic Results of PDE3A and GSK-3β

We then sought to link risk variants to candidate genes by their effect on gene expression levels or potential drug targets based on the previous bioinformatic analysis to further enhance the biological understanding of the atrial fibrillation-associated loci. We found two genes overlapping in two approaches. *PDE3A* (*p* = 4.98 × 10^−5^) and *GSK-3β* (*p* = 0.031) were first identified as upregulated DEGs between AF and sinus rhythm people in atrium tissue in the GSE2240 data set. Neither *PDE3A* nor *GSK-3β* were identified as GWAS susceptibility genes, but were included in subsequent analyses as potential AF-related genes. *PDE3A* was then identified as a potential drug target for AF. The drug active ingredients that interact with *PDE3A* are CGMP-inhibited 3′,5′-cyclic phosphodiesterase A. 

In pathway enrichment analysis, *GSK-3β* was enriched into pathways related to oxidative stress and hormone regulation, and was identified as a hub gene in the gene–gene network. The most significantly enriched pathway was the PI3K-Akt signaling pathway (Figure 7). The PI3K-Akt pathway is an intracellular signal transduction pathway that promotes metabolism, proliferation, cell survival, growth, and angiogenesis in response to extracellular signals. This is mediated through serine and/or threonine phosphorylation of a range of downstream substrates. In the pathway, AKT1 regulates the phosphorylation of the Ser9 site of GSK-3β, which leads to its inactivation. Activated PI3K converts PIP2 to PIP3. These PIPs then mop up PDK1 and Akt to the cell membrane. When PDK1 and Akt are taken to the cell membrane, Akt gets activated and phosphorylated. Overexpression of phosphatase and tensin homolog deleted on chromosome 10 (PTEN) can inhibit AKT1 phosphorylation and further activate GSK-3β. GSK-3β activity inhibits the binding of GSK-3β to BCL2 and then promotes the activation of autophagy. GSK-3β can also phosphorylate MAPK1 kinases, which is implicated in fibrogenesis.

## 4. Discussion

In this study, we applied a complementary strategy to combine the results of GWAS analysis with bioinformatics data mining in multiomics, and found that variants of potential therapeutic target *PDE3A* and key mediator gene *GSK-3β* of AF were significantly susceptible to AF in the Chinese population.

Phosphodiesterase 3A (*PDE3A)* encodes a member of the cGMP-inhibited cyclic nucleotide phosphodiesterase (cGI-PDE) family [24]. cGI-PDE enzymes hydrolyze both cAMP and cGMP, and play critical roles in many cellular processes by regulating the amplitude and duration of intracellular cyclic nucleotide signals [25]. The encoded protein mediates platelet aggregation and also plays important roles in the cardiac β-AR/AC/cAMP/PKA axis by regulating vascular smooth muscle contraction and relaxation. Our study first explored whether *PDE3A* was significantly upregulated in AF patients, which was consistent with previous research. Bernardo Dolce et al. demonstrated that in patients with persistent atrial fibrillation, the force responses to 5-HT are blunted, but they can be recovered after inhibition of *PDE3* [26]. This suggests that the change of *PDE3A* expression may cause systolic dysfunction of atrial muscle and thus participate in cardiac remodeling. Combined with the DrugBank, STITCH, and TCMSP databases, we then identified *PDE3A* as a potential drug target for AF. It was also reported that inhibitors of the encoded protein of *PDE3A* may be effective in treating congestive heart failure [27,28]. In GWAS analysis, we found that minor allele A of the intron variant *PDE3A* rs11613698 was observed to increase the risk of atrial fibrillation in the Chinese population. Although previous GWAS studies in the Chinese population did not identify PDE3A as a risk gene for AF, some indirect associations have been reported. Zun Wang et al. applied a novel metaCCA method on the GWAS summary statistics data of stroke and AF and found that *PDE3A* was a potential pleiotropic gene, which may affect ischemic or hemorrhage stroke through multiple intermediate factors such as MAPK family [29]. It was also reported by Carmen Sucharov et al. that 29 polymorphisms in the human *PDE3A* gene promoter regulated *PDE3A* gene expression, and had further effects on the electrophysiological activity of the myocardium [30]. Therefore, mutation of *PDE3A* may change the expression level of its downstream proteins, and affect the risk of atrial fibrillation by mediating cardiac remodeling.

The protein encoded by glycogen synthase kinase 3β (*GSK-3β*) is a serine-threonine kinase, belonging to the glycogen synthase kinase subfamily. It is involved in energy metabolism, neuronal cell development, and body pattern formation [31]. Numerous studies have indicated that *GSK-3β* can be phosphorylated and inhibited by protein kinase C (PKC) and then regulate a wide variety of cardiac transcription factors [32]. Recent studies suggest that calcium channel modification is a possible mediator of the association between *GSK-3β* and AF. Yan Wang et al. developed an animal model and found that ethanol can inhibit *GSK-3β* through enhanced phosphorylation, thereby leading to upregulation of T-type calcium channels (TCCs) and increased AF susceptibility [33]. In addition, it was reported that *GSK-3* can negatively regulate the sarco (endo)plasmic reticulum Ca^2+^-ATPase (SERCA) pump, a key regulator of Ca^2+^ uptake in the heart [34]. In a prospective observational study, SERCA levels in peripheral lymphocytes were reported to be associated with the outcome of pericardial ablation in patients with persistent AF, and lower levels of SERCA expression could predict the recurrence of AF after pericardial ablation [35]. Taken together, overexpression of *GSK-3β* can cause abnormal regulation of calcium ions in cardiomyocytes by inhibiting SERCA levels, leading to myocardial electrical remodeling, and thus the occurrence and maintenance of AF. 

In this study, we first found that *GSK-3β* had substantially higher levels of expression in patients with AF than in controls. Then, *GSK-3β* was identified as a hub gene in oxidative stress and cellular signaling pathways. There are several mechanisms that produce ROS in cardiac myocytes, including mitochondria, NADPH oxidase, uncoupled NO synthase, and xanthine oxidase [36,37], which increase oxidative stress and promote cardiac fibrosis. Many studies have reported that patients with AF have a decrease in antioxidant-related gene expression and an increase in ROS-related gene expression [38]. Consistent with our pathway enrichment results, previous studies have reported that regulation of oxidative stress-related gene expression was functionally associated with PI3K/AKT signaling, which is a key profibrotic element in various tissues and was reported to be capable of activating atrial fibroblasts to differentiate into myofibroblasts [39]. Several experimental studies have sought targets to inhibit atrial remodeling by affecting oxidative stress, inflammation, and the PI3K/Akt signaling pathway [40,41]. 

Our identification of the *GSK-3β* as a hub gene may help to provide a new target for the etiological study of oxidative stress pathway in AF. The minor allele G of the intron variant *GSK-3β* rs796944992 was observed to increase the risk of atrial fibrillation. Although there have been no previous reports on the *GSK-3β* variation and the risk of AF in other populations, many studies have reported that genetic alterations in *GSK-3β* are associated with various diseases mediated by oxidative stress pathways, including myocardial ischemia [42], myocardial infarction [43], and Alzheimer’s disease [44]. Our results suggest that variants of *GSK-3β* directly confer a risk of AF only in the Chinese population, indicating strong population heterogeneity in the genetics of AF. The mechanism by which *GSK-3β* mediates atrial fibrillation through the oxidative stress pathway can be further explored in future studies.

There are complex mechanisms for the occurrence and maintenance of AF, including several pathways not identified in this study, such as overinflammation and epigenetic modification. Inflammation can alter the atrial electrophysiology and structure to increase the vulnerability to AF. In a study on patients with new-onset AF, the early recurrence of AF was related to inflammatory markers, and inflammatory markers were associated with development of permanent AF [45]. However, it was reported that although oral antioxidant treatment (α-lipoic acid, ALA) reduced serum levels of common markers of inflammation in ablated patients, ALA does not prevent AF recurrence after an ablative treatment [46]. This suggests that treatment targeting inflammatory biomarkers alone may not able to revert cardiac remodeling. This may also be the reason why few genes related to inflammation were screened out as hub genes in this study. As epigenetic regulators, miRNA plays an important role in cardiac development, and the dysregulation of miRNA expression is related to cardiac remodeling. It was reported that catheter ablation was related to miRNA modulation. Several miRNAs have been reported to be able to assess and predict the risk of recurrence in patients with AF after ablation [47]. In the future, the idea of this study could be applied to miRNA bioinformatics mining, which may help to identify miRNA drug targets with clinical value after AF ablation.

We acknowledge some limitations in our study. First, three GEO data sets were included to detect DEGs in AF. There were considerable differences in their study designs hindering straightforward comparison and merging with the studies [48]. The origin of mRNAs in tissue is inconsistent, and mRNA regulation in different tissues may be contradictory. Second, although bioinformatics data mining and GWAS analysis yielded 75 hub genes and 126 loci, there were only two overlapping genes between the two methods. This may be mainly due to the fact multiomics data on AF in the Chinese population are difficult to obtain, so the genetic background of the included open data is different from our GWAS population. These factors need to be carefully considered to avoid misinterpretation of the findings. Finally, the sample size of the GWAS analysis was limited and the ratio of cases to controls was imbalanced in this study, so the main purpose of GWAS analysis was to validate AF-related genes obtained from biological information analysis in the Chinese population. This is a strategy similar to that of candidate gene association studies, and GWAS analysis itself serves as a validation function. We hope to obtain a more independent AF cohort for further analysis in future studies.

## 5. Conclusions

In conclusion, this study is the first systematic report on the screening and verification of the association of *PDE3A* and *GSK-3β* with the risk of atrial fibrillation in the Chinese population, showing that *PDE3A* is a potential drug target for AF and *GSK-3β* is a hub gene in the gene–gene network of pathways related to oxidative stress and hormone regulation. Our study provides new insights into AF mechanisms in the Chinese population.

## Figures and Tables

**Figure 1 biomedicines-11-00908-f001:**
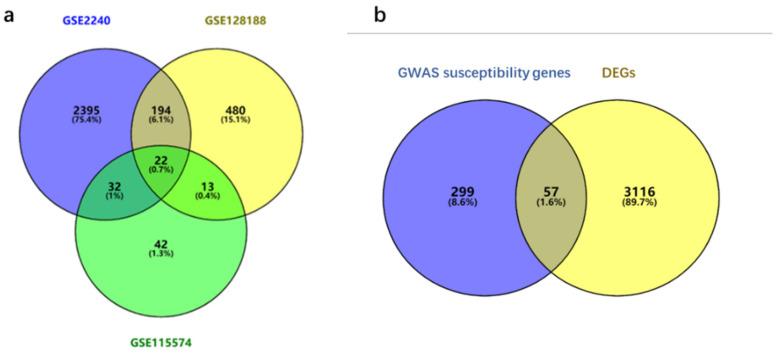
DEGs and potential AF-related genes identified from public databases. (**a**) A Venn diagram illustrating the DEG gene sets from GSE2240 (purple), GSE128188 (yellow), and GSE115574 (green) databases. A total of 3173 genes were identified as DEGs of AF, and 22 of them were overlapped in all 3 data sets. (**b**) A Venn diagram illustrating GWAS susceptibility genes (purple) and DEGs (yellow). A total of 3472 genes were identified as potential AF-related genes, and 57 of them were overlapped in both sets.

**Figure 2 biomedicines-11-00908-f002:**
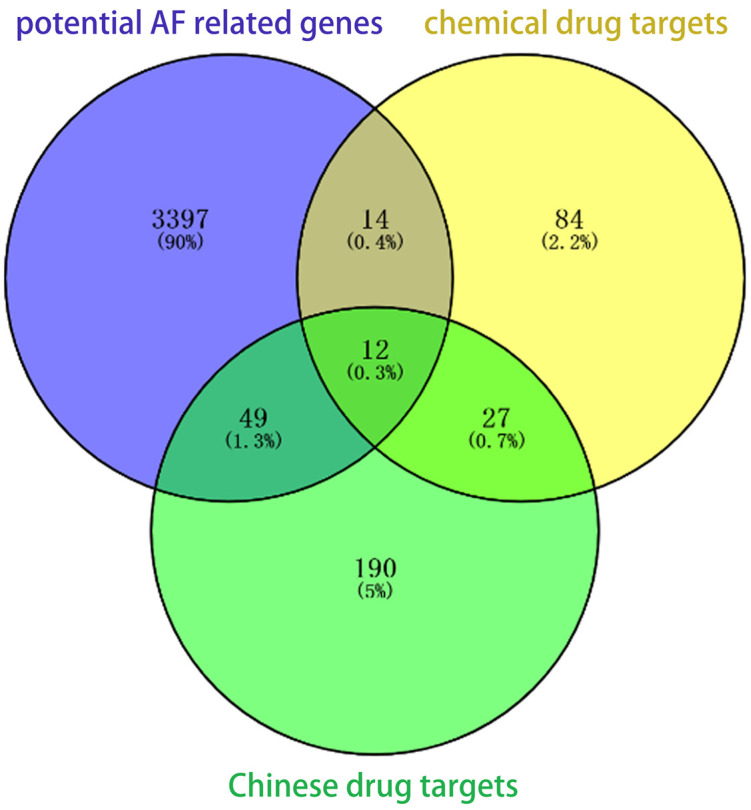
Potential therapeutic targets of AF. A Venn diagram illustrating gene sets from potential AF-related genes (purple), chemical drug targets (yellow) and Chinese drug targets (green). A total of 75 (49 + 12 + 14) AF-related genes can be used as potential therapeutic targets.

**Figure 3 biomedicines-11-00908-f003:**
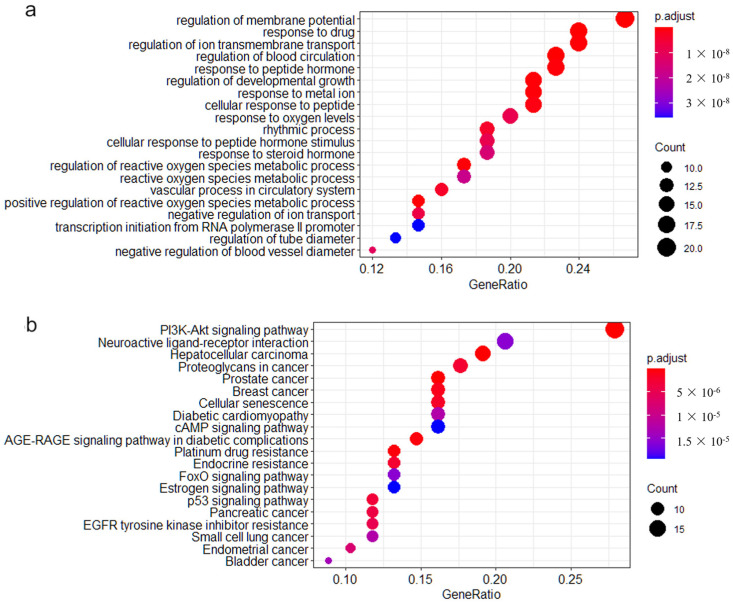
The top 20 significant biological processes and pathways of enrichment in GO and KEGG enrichment analysis. (**a**) The top 20 functionally enriched GO pathways found in the analysis of AF-related genes. (**b**) The top 20 functionally enriched KEGG pathways found in the analysis of AF-related genes. Different colors represent adjusted *p*-values. Pathways were ranked by their GeneRatio.

**Figure 4 biomedicines-11-00908-f004:**
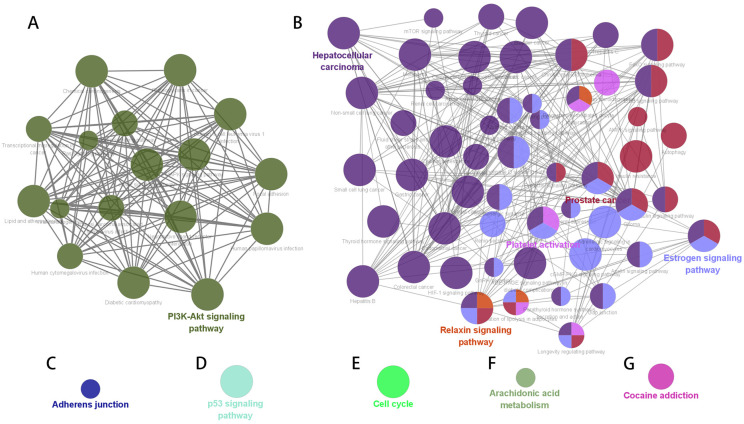
KEGG pathway enrichment network of potential AF-related genes. The node size is proportional to the enrichment significance, and the node color reflects the functional group to which it belongs. (**A**) PDK-Akt signaling pathway. (**B**) Hepatocellular carcinoma, prostate cancer, platelet activation, relaxin signaling pathway and estrogen signaling pathway. (**C**) Adherens junction. (**D**) p53 signaling pathway. (**E**) cell cycle. (**F**) Arachidonic acid metabolism. (**G**) Cocaine addiction.

**Figure 5 biomedicines-11-00908-f005:**
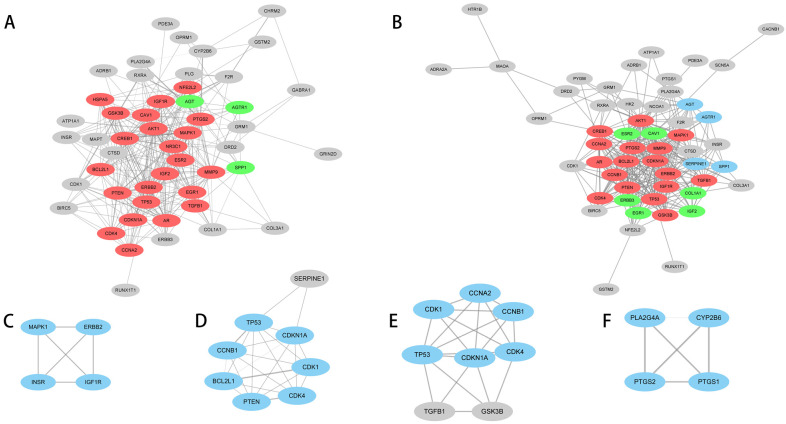
Hub genes in seven enriched pathway networks. Gray nodes are nonhub genes, color nodes are hub genes, and different color blocks represent different key gene networks under this pathway network. The (**A**–**F**) labels represent the constructed subnetworks in Table 3.

**Figure 6 biomedicines-11-00908-f006:**
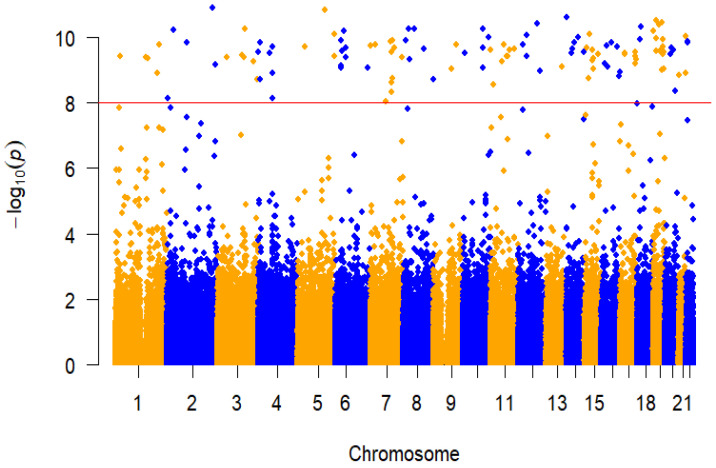
Manhattan plot of GWAS analysis of AF. The *x* axis represents the genome in physical order, and the *y* axis represents *p* values (–log10 (*p* value)) of association. The red horizontal dotted line represents a Bonferroni-corrected threshold of statistical significance corresponding to 1,000,000 independent tests (*p* < 1 × 10^−8^).

**Figure 7 biomedicines-11-00908-f007:**
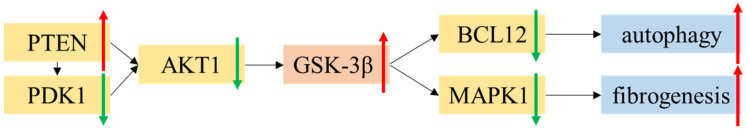
The core function of GSK-3β in PI3K-Akt signaling pathway. The upregulated proteins or processes are represented as red arrows while the downregulated proteins or processes are represented as green arrows.

**Table 1 biomedicines-11-00908-t001:** DEGs in AF identified from the GEO database.

Data Sets	Tissues	Chips (Batches)	Number of DEGs	Up-Regulated Genes	Down-Regulated Genes
GSE2240	atrium	GPL96	1819	1029	790
		GPL97	508	230	278
		GPL96 + GPL97	885	443	442
GSE128188	left auricle	GPL18573	13	3	10
	right auricle	GPL18573	1	0	1
	auricle	GPL18573	708	350	358
GSE115574	left atrium	GPL570	1	0	1
	right atrium	GPL570	1	1	0
	atrium	GPL570	111	64	47
Total			3173	1757	1432

**Table 2 biomedicines-11-00908-t002:** Potential therapeutic targets of AF.

Targets	Entrez ID	Genes
Prostaglandin G/H synthase 2	5743	PTGS2
Prostaglandin G/H synthase 1	5742	PTGS1
C-reactive protein	1401	CRP
Cellular tumor antigen p53	7157	TP53
Matrix metalloproteinase-9	4318	MMP9
Potassium voltage-gated channel subfamily H member 2	3757	KCNH2
Angiotensinogen	183	AGT
Beta-1 adrenergic receptor	153	ADRB1
Lactotransferrin	4057	LTF
Microtubule-associated protein tau	4137	MAPT
Type-1 angiotensin II receptor	185	AGTR1
RAC-alpha serine/threonine-protein kinase	207	AKT1
Alpha-2A adrenergic receptor	150	ADRA2A
Cytochrome P450 2B6	1555	CYP2B6
Sodium/potassium-transporting ATPase subunit alpha-1	476	ATP1A1
Early growth response protein 1	1958	EGR1
Sodium channel protein type 5 subunit alpha	6331	SCN5A
Triadin	10345	TRDN
Sodium channel protein type 10 subunit alpha	6336	SCN10A
Sodium channel protein type 8 subunit alpha	6334	SCN8A
Sodium channel protein type 3 subunit alpha	6328	SCN3A
Heparin cofactor 2	3053	SERPIND1
Plasminogen	5340	PLG
Voltage-dependent L-type calcium channel subunit beta-1	782	CACNB1
5-hydroxytryptamine 1B receptor	3351	HTR1B
Potassium channel subfamily K member 17	89822	KCNK17
78 kDa glucose-regulated protein	3309	HSPA5
Amine oxidase [flavin-containing] A	4128	MAOA
Androgen receptor	367	AR
Baculoviral IAP repeat-containing protein 5	332	BIRC5
Bcl-2-like protein 1	598	BCL2L1
Carbonic anhydrase II	760	CA2
Cathepsin D	1509	CTSD
Caveolin-1	857	CAV1
Cell division control protein 2 homolog	983	CDK1
Cell division protein kinase 4	1019	CDK4
CGMP-inhibited 3’,5’-cyclic phosphodiesterase A	5139	PDE3A
Collagen alpha-1(I) chain	1277	COL1A1
Collagen alpha-1(III) chain	1281	COL3A1
Cyclic AMP-responsive element-binding protein 1	1385	CREB1
Cyclin-A2	890	CCNA2
Cyclin-dependent kinase inhibitor 1	1026	CDKN1A
Cytosolic phospholipase A2	5321	PLA2G4A
D(2) dopamine receptor	1813	DRD2
DNA topoisomerase 2-alpha	7153	TOP2A
Estrogen receptor beta	2100	ESR2
G2/mitotic-specific cyclin-B1	891	CCNB1
Gamma-aminobutyric acid receptor subunit alpha-1	2554	GABRA1
Glucocorticoid receptor	2908	NR3C1
Glutamate [NMDA] receptor subunit epsilon-4	2906	GRIN2D
Glutathione S-transferase Mu 2	2946	GSTM2
Glycogen phosphorylase, muscle form	5837	PYGM
Glycogen synthase kinase-3 beta	2932	GSK-3β
Hexokinase-2	3099	HK2
Insulin-like growth factor 1 receptor	3480	IGF1R
Insulin-like growth factor II	3481	IGF2
Insulin receptor	3643	INSR
Metabotropic glutamate receptor 1	2911	GRM1
Mitogen-activated protein kinase 1	5594	MAPK1
Mu-type opioid receptor	4988	OPRM1
Muscarinic acetylcholine receptor M2	1129	CHRM2
Nuclear factor erythroid 2-related factor 2	4780	NFE2L2
Nuclear receptor coactivator 1	8648	NCOA1
Osteopontin	6696	SPP1
Phosphatidylinositol-3,4,5-trisphosphate 3-phosphatase and dual-specificity protein phosphatase PTEN	5728	PTEN
Plasminogen activator inhibitor 1	5054	SERPINE1
Protein CBFA2T1	862	RUNX1T1
Receptor tyrosine-protein kinase erbB-2	2064	ERBB2
Receptor tyrosine-protein kinase erbB-3	2065	ERBB3
Retinoic acid receptor RXR-alpha	6256	RXRA
Serum paraoxonase/arylesterase 1	5444	PON1
Thrombin	2149	F2R
Transforming growth factor beta-1	7040	TGFB1
Type I iodothyronine deiodinase	1733	DIO1
Beta-secretase 2	25825	BACE2

**Table 3 biomedicines-11-00908-t003:** Hub genes in seven enriched pathway networks.

Subnetworks	Clusters	MCODE Scores	Nodes	Lines	Hub Genes
A	1	16.818	23	185	NR3C1, TP53, ESR2, EGR1, HSPA5, CREB1, CAV1, AKT1, TGFB1, GSK-3β, MAPK1, NFE2L2, CCNA2, IGF2, BCL2L1, CDKN1A, PTGS2, PTEN, CDK4, ERBB2, MMP9, AR, IGF1R
A	2	3	3	3	AGT, SPP1, AGTR1
B	1	15.25	17	122	TP53, CREB1, AKT1, TGFB1, GSK-3β, MAPK1, CCNA2, BCL2L1, CDKN1A, PTGS2, PTEN, CCNB1, CDK4, ERBB2, MMP9, AR, IGF1R
B	2	3.333	4	5	SERPINE1, AGT, SPP1, AGTR1
B	3	3.2	6	8	ESR2, ERBB3, COL1A1, CAV1, IGF2, EGR1
C	1	4	4	6	MAPK1, ERBB2, INSR, IGF1R
D	1	7	7	21	PTEN, CDKN1A, CDK1, CCNB1, BCL2L1, TP53, CDK4
E	1	6	6	15	CDKN1A, CCNB1, CCNA2, CDK1, TP53, CDK4
F	1	4	4	6	PTGS2, PTGS1, PLA2G4A, CYP2B6
G	-	-	-	-	-

**Table 4 biomedicines-11-00908-t004:** Novel loci in GWAS analysis of AF.

Gene	SNP	Chr:position	Allele1	Allele0	MAF	β	SE	*p*
NBPF3	rs147300495	1:21777939	T	C	0.02	0.878	0.139	3.71 × 10^−10^
PDE4DIP	rs1628310	1:144868170	C	T	0.019	0.881	0.14	4.11 × 10^−10^
LOC101929703	rs974690619	1:155538990	G	A	0.019	0.881	0.14	4.54 × 10^−10^
NR5A2	rs7546336	1:199994841	C	T	0.02	0.855	0.14	1.21 × 10^−9^
SYT14	rs76437946	1:210335855	T	C	0.019	0.901	0.14	1.73 × 10^−10^
EIPR1	rs199623295	2:3359563	C	T	0.019	0.604	0.104	7.33 × 10^−9^
NLRC4	rs1408931915	2:32468262	G	A	0.019	0.916	0.139	6.05 × 10^−11^
ANKRD36C	rs5005869	2:96521297	C	T	0.019	0.903	0.14	1.50 × 10^−10^
USP37	rs182055303	2:219384724	G	A	0.019	0.955	0.14	1.26 × 10^−11^
UGT1A8	rs1042591	2:234526794	G	T	0.019	0.871	0.14	6.87 × 10^−10^
MAP4	rs137991644	3:48118180	C	A	0.02	0.877	0.139	4.16 × 10^−10^
GSK3B	rs796944992	3:119787193	G	A	0.019	0.888	0.14	3.47 × 10^−10^
LINC01565	rs141327567	3:128293330	C	A	0.02	0.876	0.139	4.07 × 10^−10^
CPNE4	rs139775204	3:131493768	T	C	0.012	0.259	0.039	5.43 × 10^−11^
TBL1XR1	rs148786696	3:176879400	G	A	0.019	0.875	0.14	5.42 × 10^−10^
MUC4	rs74500246	3:195513563	C	T	0.02	0.842	0.139	1.92 × 10^−9^
ZNF141	rs1303526299	4:382920	C	T	0.02	0.891	0.14	2.78 × 10^−10^
non-coding	rs184180522	4:8930460	A	G	0.02	0.839	0.139	1.91 × 10^−9^
non-coding	rs1394588518	4:9274164	C	A	0.02	0.894	0.139	1.49 × 10^−10^
non-coding	rs1015522933	4:55067908	G	A	0.02	0.881	0.139	3.05 × 10^−10^
UGT2B15	rs4148260	4:69531574	A	G	0.02	0.895	0.139	1.91 × 10^−10^
UGT2B15	rs3862051	4:69534405	C	T	0.02	0.857	0.14	1.20 × 10^−9^
UGT2B7	rs6600887	4:69969788	C	T	0.016	0.514	0.088	7.23 × 10^−9^
SKP2	rs763496236	5:36183898	G	A	0.02	0.895	0.139	1.91 × 10^−10^
IL4	rs376951889	5:132008827	C	T	0.02	0.699	0.103	1.44 × 10^−11^
non-coding	rs1081806	5:176198317	G	A	0.019	0.881	0.14	3.90 × 10^−10^
ADAMTS2	rs1213209228	5:178551750	G	A	0.019	0.918	0.14	7.73 × 10^−11^
MUC21	rs767391626	6:30954375	D	I	0.019	0.91	0.14	1.29 × 10^−10^
non-coding	rs1261299467	6:31030655	G	A	0.019	0.867	0.14	7.53 × 10^−10^
HLA-B	rs12697943	6:31324057	A	C	0.019	0.833	0.135	8.27 × 10^−10^
TNXB	rs200135227	6:32029369	T	C	0.02	0.884	0.139	2.60 × 10^−10^
CUL7	rs201406974	6:43014042	G	A	0.019	0.923	0.14	6.24 × 10^−11^
GSTA1	rs2894804	6:52668546	G	A	0.019	0.88	0.14	4.01 × 10^−10^
GSTA1	rs9296692	6:52668943	T	C	0.019	0.855	0.134	2.16 × 10^−10^
TULP4	rs113382463	6:158847210	G	A	0.02	0.858	0.139	8.69 × 10^−10^
non-coding	rs1443575050	7:2919444	G	A	0.019	0.901	0.14	1.78 × 10^−10^
IGF2BP3	rs118111412	7:23416772	C	T	0.02	0.893	0.139	1.67 × 10^−10^
non-coding	rs1158528647	7:75813735	C	T	0.016	0.51	0.088	9.18 × 10^−9^
CYP3A5	rs76293380	7:99250394	D	I	0.019	0.908	0.14	1.39 × 10^−10^
MUC3A	rs73163748	7:100549573	T	G	0.019	0.892	0.14	2.83 × 10^−10^
MUC3A	rs74197937	7:100550133	G	A	0.017	0.706	0.12	4.75 × 10^−9^
MUC12	rs75466554	7:100615387	C	T	0.02	0.837	0.139	2.39 × 10^−9^
KMT2E	rs149680168	7:104698096	C	T	0.02	0.902	0.139	1.25 × 10^−10^
LAMB1	rs6959803	7:107621620	A	G	0.02	0.845	0.139	1.75 × 10^−9^
CFTR	rs34517638	7:117146425	C	T	0.02	0.891	0.139	2.14 × 10^−10^
KMT2C	rs74977767	7:151932876	T	G	0.019	0.849	0.135	4.14 × 10^−10^
non-coding	rs5004426	8:13679081	C	T	0.02	0.896	0.138	1.28 × 10^−10^
non-coding	rs375571715	8:24929402	A	G	0.02	0.917	0.139	5.65 × 10^−11^
non-coding	rs149373333	8:29332974	T	C	0.02	0.875	0.139	4.84 × 10^−10^
RP1	rs759385909	8:55533903	T	C	0.02	0.922	0.139	5.41 × 10^−11^
RIDA	rs187976179	8:99119874	A	G	0.02	0.867	0.136	2.33 × 10^−10^
ARHGAP39	rs113552609	8:145790524	C	T	0.02	0.842	0.139	1.93 × 10^−9^
SLC28A3	rs112132735	9:86980920	G	A	0.02	0.86	0.139	9.18 × 10^−10^
LOC105376205	rs12553896	9:110375131	C	A	0.019	0.905	0.141	1.73 × 10^−10^
non-coding	rs190890438	10:11501497	C	T	0.019	0.89	0.14	2.97 × 10^−10^
CYP2C19	rs540418228	10:96561899	G	A	0.02	0.89	0.139	2.17 × 10^−10^
CYP2C19	rs113934938	10:96602752	G	A	0.019	0.917	0.139	5.70 × 10^−11^
CYP2C9	rs774607211	10:96701973	G	A	0.019	0.865	0.14	8.46 × 10^−10^
non-coding	rs199864119	10:127201227	A	G	0.02	0.905	0.139	1.01 × 10^−10^
MUC6	rs200695483	11:1016957	G	T	0.019	0.897	0.141	2.51 × 10^−10^
BTBD10	rs1399839941	11:13436715	T	C	0.017	0.715	0.119	2.76 × 10^−9^
OR9G1	rs78036340	11:56467819	G	A	0.02	0.875	0.14	5.37 × 10^−10^
non-coding	rs188394047	11:62423576	A	G	0.02	0.889	0.138	1.69 × 10^−10^
non-coding	rs191739873	11:79947826	C	A	0.019	0.883	0.14	3.87 × 10^−10^
non-coding	rs375180756	11:81527364	T	C	0.02	0.876	0.139	3.85 × 10^−10^
non-coding	rs200877836	11:89331149	G	A	0.02	0.89	0.14	2.50 × 10^−10^
non-coding	rs76168892	11:113838928	C	T	0.019	0.896	0.14	2.24 × 10^−10^
PDE3A	rs11613698	12:20743447	A	G	0.019	0.898	0.14	1.72 × 10^−10^
non-coding	rs1158482043	12:38739656	A	G	0.019	0.918	0.14	8.31 × 10^−11^
non-coding	rs796791101	12:38739890	C	T	0.019	0.882	0.14	3.88 × 10^−10^
non-coding	rs10777371	12:92375102	G	A	0.019	0.931	0.14	3.95 × 10^−11^
TDG	rs372823872	12:104359779	C	T	0.02	0.857	0.14	1.10 × 10^−9^
HS6ST3	rs117321325	13:97247454	T	C	0.02	0.86	0.139	7.71 × 10^−10^
AP1G2	rs199900328	14:24031771	G	A	0.02	0.943	0.14	2.43 × 10^−11^
non-coding	rs915392393	14:46302004	T	C	0.02	0.879	0.138	3.02 × 10^−10^
LINC01599	rs56246200	14:50523448	G	A	0.019	0.899	0.141	2.25 × 10^−10^
non-coding	rs141410596	14:59276647	C	A	0.02	0.897	0.139	1.42 × 10^−10^
NRXN3	rs117501552	14:79223516	T	C	0.02	0.904	0.139	1.02 × 10^−10^
non-coding	rs2919632	14:105325224	G	A	0.019	0.893	0.14	2.78 × 10^−10^
LOC100288637	rs12902692	15:31042778	C	T	0.019	0.898	0.14	2.03 × 10^−10^
non-coding	rs181979589	15:38936842	C	T	0.02	0.838	0.138	1.76 × 10^−9^
SORD2P	rs2462045	15:45138196	C	T	0.019	0.917	0.14	8.08 × 10^−11^
ALDH1A2	rs151009495	15:58318627	A	G	0.02	0.872	0.139	5.21 × 10^−10^
AQP9	rs59710194	15:58470240	A	G	0.02	0.887	0.139	2.51 × 10^−10^
GOLGA2P11	rs1188201439	15:62538744	G	A	0.019	0.863	0.14	9.46 × 10^−10^
SLC24A1	rs182375570	15:65920491	A	C	0.02	0.868	0.138	4.54 × 10^−10^
non-coding	rs76141469	15:85841674	A	G	0.019	0.88	0.139	3.23 × 10^−10^
ABCC1	rs779233080	16:16117841	C	T	0.02	0.871	0.14	6.38 × 10^−10^
non-coding	rs651252	16:26733084	C	T	0.019	0.901	0.14	1.77 × 10^−10^
SH2B1	rs79172792	16:28872240	C	T	0.02	0.866	0.14	7.85 × 10^−10^
ABCC12	rs9302750	16:48149467	G	A	0.019	0.896	0.139	1.44 × 10^−10^
non-coding	rs147492577	16:78128820	C	T	0.019	0.899	0.14	1.96 × 10^−10^
ZCCHC14-DT	rs1027308723	16:87538248	C	T	0.017	0.592	0.097	1.57 × 10^−9^
ANKRD11	rs1345338837	16:89530633	T	C	0.019	0.852	0.139	1.18 × 10^−9^
MAP2K3	rs56369732	17:21215552	T	C	0.02	0.886	0.14	3.35 × 10^−10^
KCNJ12	rs76267885	17:21319743	G	A	0.02	0.887	0.14	2.95 × 10^−10^
non-coding	rs2620043	17:71833095	G	A	0.019	0.887	0.14	3.74 × 10^−10^
non-coding	rs1229451307	17:73434022	G	A	0.019	0.848	0.136	6.57 × 10^−10^
UNC13D	rs145811063	17:73833216	G	A	0.02	0.883	0.139	2.81 × 10^−10^
non-coding	rs9902558	17:76635886	G	A	0.019	0.881	0.14	4.59 × 10^−10^
ANKRD20A5P	rs200431864	18:14179412	G	A	0.019	0.909	0.14	1.20 × 10^−10^
OSBPL1A	rs4800569	18:21962607	C	T	0.02	0.915	0.138	4.84 × 10^−11^
non-coding	rs140152353	19:2379463	G	A	0.02	0.921	0.14	6.35 × 10^−11^
CYP4F12	rs149121004	19:15792430	G	A	0.019	0.895	0.141	2.59 × 10^−10^
CYP4F2	rs4020346	19:15989730	C	T	0.02	0.908	0.14	1.11 × 10^−10^
MED26	rs75838150	19:16691304	C	T	0.019	0.935	0.139	2.99 × 10^−11^
non-coding	rs181407642	19:30211111	C	A	0.02	0.915	0.138	4.26 × 10^−11^
CHST8	rs4805919	19:34148544	C	T	0.02	0.879	0.138	2.69 × 10^−10^
CYP2S1	rs187503524	19:41708962	A	G	0.019	0.854	0.139	9.88 × 10^−10^
RELB	rs187976859	19:45533628	C	A	0.02	0.925	0.138	3.50 × 10^−11^
non-coding	rs113900465	19:46748849	G	A	0.019	0.893	0.14	2.79 × 10^−10^
CCDC9	rs144604956	19:47766123	G	A	0.019	0.894	0.14	2.72 × 10^−10^
SULT2B1	rs7248627	19:49084356	C	T	0.019	0.89	0.14	3.25 × 10^−10^
non-coding	rs73046773	19:49582674	G	A	0.02	0.904	0.139	1.18 × 10^−10^
SIGLEC11	rs375426790	19:50451606	G	A	0.02	0.897	0.14	1.92 × 10^−10^
ZNF350	rs113541493	19:52484200	G	A	0.02	0.886	0.139	2.74 × 10^−10^
CACNG8	rs1280762104	19:54478295	C	T	0.02	0.858	0.139	9.13 × 10^−10^
LINC01733	rs73601743	20:25947923	C	T	0.019	0.888	0.14	3.28 × 10^−10^
FRG1BP	rs138922778	20:29632727	D	I	0.019	0.87	0.136	2.08 × 10^−10^
ZHX3	rs1363587961	20:39898770	G	A	0.019	0.894	0.14	2.35 × 10^−10^
SNAI1	rs749239193	20:48600541	C	T	0.019	0.612	0.104	4.32 × 10^−9^
non-coding	rs113148353	21:14361624	C	T	0.019	0.834	0.137	1.40 × 10^−9^
non-coding	rs79163003	21:47475650	A	G	0.02	0.85	0.139	1.25 × 10^−9^
DIP2A	rs368500330	21:47986571	C	T	0.019	0.913	0.14	9.18 × 10^−11^
non-coding	rs61731379	22:22730619	A	T	0.019	0.904	0.14	1.33 × 10^−10^
non-coding	rs67984407	22:24230687	C	T	0.019	0.907	0.14	1.45 × 10^−10^

## Data Availability

The data presented in this study are available on request from the corresponding author. The data are not publicly available due to the policy of the Ethics Committee of the Peking University Health Science Center.

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
