# Peer review of "PDE3A and GSK3B as Atrial Fibrillation Susceptibility Genes in the Chinese Population via Bioinformatics and Genome-Wide Association Analysis"

_biomedicines, 2023, doi:10.3390/biomedicines11030908_

Round 1

Reviewer 1 Report (Previous Reviewer 1)

I have no further questions

Author Response

We gratefully appreciate for your review.

Reviewer 2 Report (New Reviewer)

The paper promises more in the introduction than is later presented. The authors promise to present the metabolic pathways involved in the malfunction of particular genes. There is no such analysis that demonstrates the step-by-step flow of metabolites along metabolic pathways, their inhibition or activation and the effect of these regulations ultimately on the onset of disease. Please provide at least one such metabolic pathway from beginning to end. Mentioning SNPs alone does not distinguish this manuscript from others of its kind. Also missing is a discussion of the consequences of metabolic pathway regulation with other authors. In addition, the study is with Chinese population cases. Although numerous, there is a lack of clear extrapolation and explanation of what the relationship of the results of these studies might be for other genetic populations.

The paper is with the author's correction not removed.

Author Response

Reviewer 3 Report (New Reviewer)

The authors provide the results of the study of GWAS and DEG combination in Chinese population. The main problem is the sample size, 110 cases have limited power for GWAS in atrial fibrilation, moreover for DEG with very few sample data, although the authors identified significant associations. This limitation shoud be discussed and compare with previously published studies.

Some additional information regarding atrial fibrilation condition should be included: first or recurrent atrial fibrilation, previous medical intervention.

The conclusions focus on PDE3A and GSK3B, genes already known to be involved in atrial fibrilation in several studies. Therefore, the study should be focused on validation in Chinese population

Author Response

This manuscript is a resubmission of an earlier submission. The following is a list of the peer review reports and author responses from that submission.

Round 1

Reviewer 1 Report

In this manuscript, Zhou et.al. screened several pivotal genes involved in atrial fibrillation pathogenesis and identified PDE3A and GSK3B as  susceptibility genes of AF in Chinese population by analyzing the public datasets. Although interesting results have been shown, this study could be potentially improved as follow:
1. How the 110 AF cases and 1201 controls were selected in GWAS study? as the control numbers are much greater than AF cases, which may cause the imbalanced power in statistical analysis.

2.GSE128188 has very few DEGs in left or right auricles, but much higher number of auricle. What's the difference of these three samples? How to explain the limited number of DEGs in the first two samples.

3. Although the GWAS genes are overlapped with DEGs in the datasets, have the authors tried to intersect the GWAS SNPs with the DEGs in a limited window, i.e 10kb ? or calculated the linkage disequilibrium of GWAS lead SNPs with novel loci of table4?

4. It will be great to label the Fangshan and adjacent populations on Fig.6 

Author Response

  1. How the 110 AF cases and 1201 controls were selected in GWAS study? as the control numbers are much greater than AF cases, which may cause the imbalanced power in statistical analysis.

Response 1: We appreciate the question and comments very much. The population was recruited from the Fangshan Family‐based Study in China, a community-based and hospital-centered genetic epidemiological study of chronic cardiovascular, cerebrovascular and metabolic diseases. Cases with confirmed AF were included as probands; after their informed consent is obtained, their parents, siblings, or spouses are recruited and screened by using the proband-initiated contact method. AF was verified at the central hospital (the First Hospital of Fangshan District, Beijing, China). Controls were selected from relatives of the AF probands and other chronic patients of similar age and sex in the program. We acknowledge that the ratio of cases to controls in this study was imbalanced due to the limited number of AF patients we collected and genotyped. We chose as many controls as possible to ensure the total sample size. Even so, the proportion of cases in this study is similar to the proportion of AF patients in middle-aged and elderly people in China, which can better reflect the genetic characteristics of  AF patients in the natural population. Besides, we have adjusted the first ten principal components of the genome in our analysis, which could reduce the bias and false positives caused by population structure to some extent.

2.GSE128188 has very few DEGs in left or right auricles, but much higher number of auricle. What's the difference of these three samples? How to explain the limited number of DEGs in the first two samples.

Response 2: Thank you for your question. The sample of GSE128188 consists of auricle tissues from 10 AF patients and 10 people with sinus rhythm. There were 5 cases of left auricles and 5 cases of right auricles in each group. GSE2240 and GSE115574 consist of atrium tissues. We observed that, in contrast to the results of all tissues, only a small amount of DEGs were identified in auricles and atriums when stratified analysis of left and right tissues was performed. The purpose of stratified analysis was to explore whether there is specificity of left and right tissues in DEGs. However, the results the results showed no tissue specificity. We thought that this may also be because the tissue sample size in stratified analysis was only half of that in combined analysis, which could not reach the statistical power of the LIMMA method.

  1. Although the GWAS genes are overlapped with DEGs in the datasets, have the authors tried to intersect the GWAS SNPs with the DEGs in a limited window, i.e 10kb ? or calculated the linkage disequilibrium of GWAS lead SNPs with novel loci of table4?

Response 3: Thank you for your comments. We have added LocusZoom plots of novel loci of table4 in Figure S2 of supplement, which show the linkage disequilibrium around the PDE3A rs11613698 and GSK3B rs796944992.

  1. It will be great to label the Fangshan and adjacent populations on Fig.6

Response 4: Thank you for your comments. We have refined Figure 6, and marked Fangshan and East Asian people in the picture. At the suggestion of another reviewer, we have put Figure 6 in the supplement and renamed it Figure S1.

Reviewer 2 Report

In this study, the authors performed the bioinformatic analysis to identify atrial fibrillation (AF) susceptible genes in Chinese population. The authors first compiled a large list of AF related genes using differential expression analysis, public GWAS susceptibility genes, therapeutic targets analysis, and pathway analysis. The authors then performed the GWAS study using 110 AF cases and 1201 controls to identify significant AF risk loci. They found that PDE3A and GSK3B loci overlapped with the AF-related gene list and suggested that they might be AF risk genes in Chinese population. Even though the authors have performed a wide range of bioinformatic analyses, most analyses are not convincing. The authors used a very loose criteria to compile the AF gene list and has shown little evidence that those genes are real AF risk genes. The GWAS analysis is limited by its small cohort size and lack of replication cohort. The authors also didn’t do any comparison of their results with existing results for AF GWAS studies in Chinese or other populations. Overall, the study design is poor and the results are not convincing.

My specific comments are as below:

1.     The definition of potential AF related genes is too loose. The authors should take the overlapped 22 genes from the three expression datasets and compare them with GWAS susceptible genes, and then also take the overlapped ones.

2.     Line 195: For pathway enrichment analysis, the authors should indicate what genes are included in the analysis.

3.     For figures, figure legend should be added to explain each panel.

4.     Figure 6 should be in the supplement.

5.     The Manhattan plot of GWAS study doesn’t look convincing. The authors should replicate the analysis in a separate cohort.

6.     The authors should discuss the GWAS findings with the other related studies, such as GWAS study of AF in Chinese population and GWAS of AF in other populations.

7.     Table 4: The authors should annotate the table with nearby gene names.

8.     Line 244-247: More information about PDE3A and GSK-3beta from the first approach should be provided. For instance, which pathway network do they belong to? Are they differentially expressed in AF samples? Are they GWAS susceptible genes?

Author Response

  1. The definition of potential AF related genes is too loose. The authors should take the overlapped 22 genes from the three expression datasets and compare them with GWAS susceptible genes, and then also take the overlapped ones.

Response 1: We appreciate the question and comments very much. We acknowledge that the more rigorous criteria for incorporating the intersection of three DEGs data sets into subsequent analyses would have made the results more convincing. However, the purpose of our analysis of DEGs in the first step was to identify as many genes as possible that are potentially associated with the occurrence of AF. The three GEO datasets were collected from people with different genetic backgrounds and different myocardial tissues, so the DEGs among them were difficult to compare directly. Therefore, we thought that DEGs identified in only one or two datasets cannot rule out its potential association with AF, which needs to be verified in combination with subsequent bioinformation analysis and population studies

  1. Line 195: For pathway enrichment analysis, the authors should indicate what genes are included in the analysis.

Response 2: Thank you for your advice. We added to the beginning of this paragraph that Pathway enrichment analysis were performed on 75 AF related genes above.

  1. For figures, figure legend should be added to explain each panel.

Response 3: Thank you for your comments. We have added the legends of all the figures.

  1. Figure 6 should be in the supplement.

Response 4: Thank you for your comments. We have put Figure 6 in the supplement and renamed it Figure S1.

  1. The Manhattan plot of GWAS study doesn’t look convincing. The authors should replicate the analysis in a separate cohort.

Response 5: Thank you for your comments. We acknowledge that the sample size of the GWAS analysis in this study was not large and we were not able to get an accessible validation cohort for AF in the Chinese population. However, our study design differs from traditional GWASs studies. Traditional GWASs analysis is not based on any prior information, and its aim is primarily to identify new susceptibility SNPs in the population. In this study, the main purpose of GWAS analysis was to validate AF related genes obtained from biological information analysis in the Chinese population. This is a strategy similar to that of candidate gene association studies, and GWAS analysis itself serves as a validation function. We hope to obtain an independent AF cohort for further analysis in future studies

  1. The authors should discuss the GWAS findings with the other related studies, such as GWAS study of AF in Chinese population and GWAS of AF in other populations.

Response 6: Thank you for your comments. Previous GWAS studies carried out in the Chinese population have not identified PDE3A or GSK-3β as risk genes for AF. This may be due to a lack of studies with large enough samples in China. We have supplemented the discussion section with studies based on GWAS summary statistics or experiments on the genes identified in this study, which reported that these genes show pleiotropic associations with AF and its long-term outcomes such as stroke or heart failure. These studies illustrate the complexity of the association of these genes with AF.

  1. Table 4: The authors should annotate the table with nearby gene names.

Response 7: Thank you for your comments. We have added the gene names in Table 4. In addition, we have updated the SNP name to standard rsIDs.

  1. Line 244-247: More information about PDE3A and GSK-3beta from the first approach should be provided. For instance, which pathway network do they belong to? Are they differentially expressed in AF samples? Are they GWAS susceptible genes?

Response 8: Thank you for this very valuable advice. We have re-written this part according to your suggestion to detail the results of each step in the bioinformatics analysis of the two genes.

Round 2

Reviewer 1 Report

The authors' responses released my concern and I have no further questions for the revised manuscript.

Reviewer 2 Report

Even though the authors tried to address all the questions, they still didn’t provide direct answers some key questions. For instance, the authors didn’t change their strategies to prioritize potential AF related genes, neither did they refine their GWAS analysis. The overall analyses and results of the manuscript are still not convincing.